# Reliable cell cycle commitment in budding yeast is ensured by signal integration

**Xili Liu**[1,2†‡], **Xin Wang**[1,2†], **Xiaojing Yang**[1,2], **Sen Liu**[3], **Lingli Jiang**[1,2], **Yimiao Qu**[1,2], **Lufeng Hu**[1,2], **Qi Ouyang**[1,2], **Chao Tang**[1,2]*

[1]Center for Quantitative Biology, Peking University, Beijing, China; [2]Peking-Tsinghua Center for Life Sciences, Peking University, Beijing, China; [3]Institute of Molecular Biology, College of Medical Science, China Three Gorges University, Yichang, China

**Abstract** Cell fate decisions are critical for life, yet little is known about how their reliability is achieved when signals are noisy and fluctuating with time. In this study, we show that in budding yeast, the decision of cell cycle commitment (Start) is determined by the time integration of its triggering signal Cln3. We further identify the Start repressor, Whi5, as the integrator. The instantaneous kinase activity of Cln3-Cdk1 is recorded over time on the phosphorylated Whi5, and the decision is made only when phosphorylated Whi5 reaches a threshold. Cells adjust the threshold by modulating Whi5 concentration in different nutrient conditions to coordinate growth and division. Our work shows that the strategy of signal integration, which was previously found in decision-making behaviors of animals, is adopted at the cellular level to reduce noise and minimize uncertainty.

*For correspondence: tangc@pku.edu.cn

†These authors contributed equally to this work

**Present address:** ‡Department of Systems Biology, Harvard University, Boston, United States

**Competing interests:** The authors declare that no competing interests exist.

## Introduction

Extensive studies have shown the importance of precise cell fate decisions in many life activities, such as cell cycle entry in response to environmental changes and pattern formation during embryonic development (*Xiong and Ferrell, 2003*; *Gregor et al., 2007*; *Balázsi et al., 2011*). However, little is known about how cells utilize the information of the input signal to make robust and reliable decisions especially in cases of noisy and time-varying signals. We address this issue by using the Start transition in budding yeast (*Saccharomyces cerevisiae*) as a model system. Start is a major cell cycle checkpoint in budding yeast (corresponding to the restriction point in mammalian cells), which decides whether or not the cell should make the irreversible commitment to the next round of division (*Hartwell et al., 1974*). The environmental and internal conditions are sensed and passed to the Start signal Cln3 (*Gallego et al., 1997*; *Polymenis and Schmidt, 1997*; *Hall et al., 1998*; *Parviz et al., 1998*; *Menoyo et al., 2013*). As a G1 cyclin, Cln3 triggers the Start transition by activating a downstream positive feedback loop composed of the repressor Whi5, the transcription factor SBF/MBF and the cyclin Cln1/2 (*Skotheim et al., 2008*; *Charvin et al., 2010*) (*Figure 1A* upper panel). The Start checkpoint coordinates cell growth and cell division and is thought to control the cell size under different growth conditions (*Johnston et al., 1977*; *Jorgensen and Tyers, 2004*). However, the mechanisms of the coordination and control have not been fully elucidated.

It was proposed that Cln3 concentration in the nucleus (or total Cln3 abundance) increases with cell size and the Start transition is triggered when Cln3 level reaches a critical threshold (*Chen et al., 2004*; *Jorgensen and Tyers, 2004*). There are several issues with this model. For instance, it is unclear how nuclear Cln3 concentration is coupled to cell size. Furthermore, both the *CLN3* mRNA and the Cln3 protein turn over very fast (with a few minutes half-lives [*Cross and Blake, 1993*; *Tyers et al., 1992*; *Yaglom et al., 1995*]) at very low abundance (a few copies of the mRNA [*McInerny et al., 1997*; *Arava et al., 2003*] and 100–200 copies of the protein [*Tyers et al., 1993*]), enabling it to rapidly respond to the environmental and cellular condition changes. This would imply considerable noise and fluctuations

**eLife digest** Budding yeast and other single-celled organisms can reproduce by dividing to produce two daughter cells. The timing of the cell division is critical because if the cell is still small when it divides, the resulting daughter cells may not be big enough to survive.

In budding yeast, the irreversible decision to divide—known as the 'Start' checkpoint—is only made once a cell reaches a certain size and is triggered by a protein called Cln3. This protein controls the activity of another protein called Whi5, which normally prevents the cell from dividing by switching off particular genes. Cln3 adds phosphate groups to Whi5 to make 'phosphorylated Whi5', which allows the genes involved in cell division to be switched on.

It is commonly believed that the level of Cln3 reflects the size of the cell and the nutrient conditions. Therefore, one model of cell division proposes that the cell passes the Start checkpoint when the level of Cln3 reaches a threshold value. However, levels of the Cln3 protein in cells can naturally fluctuate, and computer simulations based on this model showed that this would not produce reliable decisions on when to divide. So how do cells manage to distinguish noise from the genuine signals that indicate it is the right time to divide?

To address this question, Liu et al. studied yeast cells containing an artificial version of the gene encoding the Cln3 protein whose levels could be adjusted by adding a particular chemical. This revealed that cells with higher levels of Cln3 passed through the Start checkpoint sooner than cells that had lower levels of Cln3.

The observation suggests that cells add up the amount of Cln3 present over a period of time to see if this reaches the threshold needed for the Start checkpoint. This could be possible if, instead of sensing Cln3 levels directly, the cell senses the accumulation of phosphorylated Whi5. To test this idea, Liu et al. carried out additional experiments and found that the decision to pass the Start checkpoint only occurs when the amount of phosphorylated Whi5 reaches a certain threshold.

The cells are able to coordinate their growth and division under different nutrient conditions by altering the threshold of phosphorylated Whi5. When the nutrient supply is poor, more phosphorylated Whi5 needs to be accumulated to allow the cell to pass the Start checkpoint. In this way, cells adjust when they divide according to nutrient conditions. Similar strategies may be found in other signaling or decision-making systems.

in the Cln3 profile. If the Start transition were triggered by the instantaneous Cln3 concentration passing above a threshold (*Chen et al., 2004*; *Jorgensen and Tyers, 2004*), the decision would be rather stochastic and could be unreliable.

In this study, by quantitatively measuring an inducible Cln3 mutant and simultaneously monitoring the timing of Start transition in single cells, we address the question of what information in the Cln3 profile the cell is using to make the Start decision. We found that the Start transition is triggered by the time integration of Cln3 activity on phosphorylated Whi5. Furthermore, cells modulate Whi5 concentration in different nutrient conditions. Cln3 and Whi5 together control G1 length through the time integration mechanism to coordinate cell division with cell growth. The time integration strategy can reduce noise and minimize uncertainty in the decision and may be widely implied in decision making systems at the cellular level.

## Results

### G1 length is inversely proportional to Cln3 concentration

To quantitatively investigate how Start transition is triggered by Cln3, we first integrated a copy of inducible *CLN3* onto the genome in a strain lacking the native *CLN3* and *BCK2* and fused the endogenous *WHI5* with the red fluorescent protein tdTomato. The G1 length $T_{G1}$ is defined as the time interval between Whi5 nuclear entry in late mitosis and its exclusion from the nucleus at the Start transition (*Taberner et al., 2009*) (*Figure 1B* and *Figure 1—figure supplement 1*), which is a measure for how long the cell waits to make the Start decision. Cln3 level under a synthetic inducible promoter *GlacSpr* (*Figure 1—figure supplement 2*) was controlled by titrating the inducer IPTG. The cells were grown

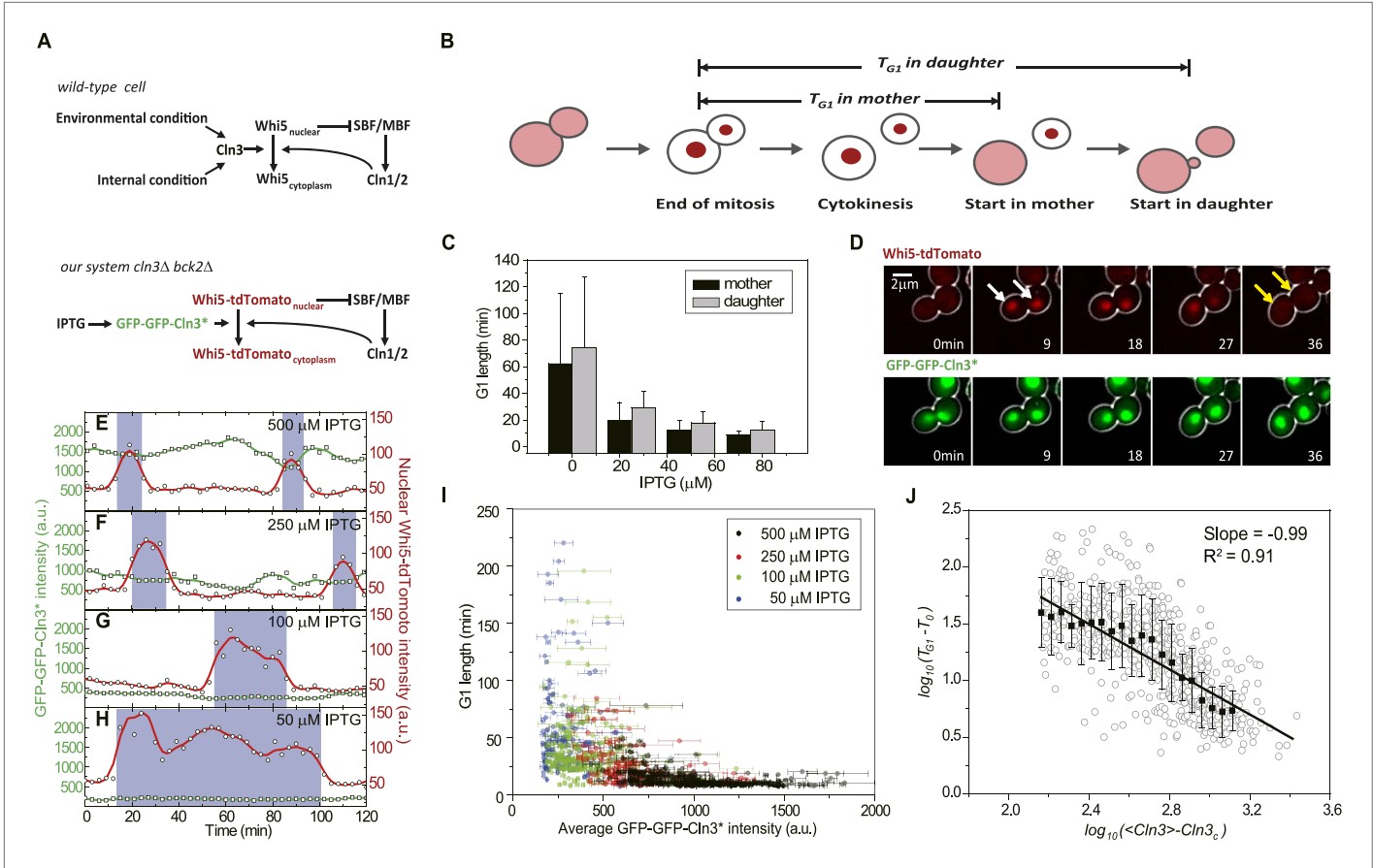

**Figure 1**. G1 length is inversely proportional to the average Cln3. (**A**) Schematic of the Start regulatory network in wild-type (upper panel) and in strains used in this study (lower panel). (**B**) G1 length is defined as the time interval during which Whi5 resides in the nucleus (*Figure 1—figure supplement 1*). Whi5 localization is schematically shown in red. (**C**) Population-averaged G1 length at different IPTG concentrations for cells carrying IPTG-induced *CLN3* as the Start signal (Strain YCT2002). Error bars represent standard deviation. (**D**) Composite bright-field and fluorescence images for cells carrying *GFP-GFP-CLN3** and *WHI5-tdTomato* (Strain YCT2003) under full induction (2 mM IPTG). White arrows indicate the beginning of G1; yellow arrows indicate the timing of Start transition. (**E**–**H**) Time courses of the GFP-GFP-Cln3* (green) and nuclear Whi5-tdTomato (red) fluorescent intensities in a representative single cell at each different IPTG concentration. The empty squares and circles denote the raw data of Cln3 and nuclear Whi5 fluorescence, respectively; the green and red lines are the smoothing splines of the raw data; purple boxes show the G1 duration. (**I**) G1 length vs the average Cln3 fluorescent intensity in G1. Each dot represents a measurement from one cell cycle event; error bars indicate standard deviation. Color groups represent data collected from each of the corresponding IPTG concentrations. (**J**) G1 length vs the average Cln3 fluorescence in log–log scale (empty circles are calculated from **I**). <Cln3> represents the average Cln3 fluorescence in G1; $T_0$ and $Cln3_c$ are fitted from **I** as described in 'Materials and methods'. The solid line is the best linear fit of the binned data (filled squares). Error bars indicate standard deviation.

The following figure supplements are available for figure 1:

**Figure supplement 1**. Definition of G1 length.

**Figure supplement 2**. The promoter *GlacSpr* is repressed by LacI and induced by IPTG.

**Figure supplement 3**. The homologous structure of Cln3-Cdk1 complex.

**Figure supplement 4**. Cell size and GFP brightness of selected Cln3 mutants under full induction.

**Figure supplement 5**. Size control is maintained with the mutant Cln3, Cln3*.

**Figure supplement 6**. The correlation between G1 length and average Cln3 fluorescence intensity in mother and daughter cells (YCT2003).

*Figure 1. Continued on next page*

*Figure 1. Continued*

**Figure supplement 7**. The correlation between G1 length and average Cln3 fluorescence intensity with different Cln3 signals: *CLNR108A* (YCT2003) and *CLN3D166A* (YCT2004).

**Figure supplement 8**. The correlation between G1 length and average Cln3 fluorescence intensity with different G1 length markers: WHI5-tdTomato (YCT2003) and MCM-mCherry (YCT2010).

**Figure supplement 9**. G1 length is inversely proportional to averaged Cln3 intensity without deducting asymptotes.

in a microfluidic chip and monitored by time-lapse microscopy. We found that in both mother and daughter cells G1 length is prolonged as IPTG concentration decreases, which is consistent with the previous findings that increased Cln3 dosage shortens G1 length (*Di Talia et al., 2007*) and suggests a negative correlation between G1 length and Cln3 level (*Figure 1C*).

The low abundance and short half-life of the wild-type Cln3 make its detection in single cells extremely difficult (data not shown). To better quantitate the observed negative correlation between G1 length and Cln3 level, we screened for Cln3 mutants with lower activity and longer half-life based on a homolog modeling of the Cln3-Cdk1 complex (*Figure 1—figure supplement 3*). Lower activity requires higher Cln3 concentration to pass Start and longer half-life allows more fluorescent proteins to mature before the tagged Cln3 is degraded, thus making the mutant Cln3 detectable in single cells. The fluorescent intensity and CDK activity of the GFP-GFP-Cln3 mutants were carefully examined. The desired mutants should fulfill three criteria: 1) the fluorescent brightness of the mutant can be quantitatively measured; 2) under full induction, the mutant can rescue the physiological function of the endogenous Cln3 in *cln3Δ* cells (*Figure 1—figure supplement 4*); 3) the cell cycle of *cln3Δ bck2Δ* strain can be arrested when the mutant is shut off. Two successful mutants, *CLN3R108A* and *CLN3D166A*, were obtained from the screening. Since the fully induced mutants do not change cell cycle behaviors such as the doubling time and cell size, we consider them faithful replacements of wild-type *CLN3* under our experimental conditions. Unless otherwise specified, the mutant *CLN3R108A* (denoted *CLN3\**) was used in following experiments (*Figure 1A* lower panel and D). Interestingly, we found that under full induction, the nuclear concentration of GFP-GFP-Cln3* is positively correlated and G1 length is negatively correlated with birth size (*Figure 1—figure supplement 5*), which implies that size control is maintained with the mutant Cln3 (*Di Talia et al., 2007*). It was proposed that Cln3 responds to size through the upstream open reading frame (uORF) of its mRNA (*Polymenis and Schmidt, 1997*) and/or ER-associated proteins Whi3 and Ydj1 (*Wang et al., 2004*; *Vergés et al., 2007*). However, in our construct, the 5′ leader of Cln3 mRNA was removed, and we did not observe any ER-like localization of Cln3 protein. The promoter activity does not correlate with size either (data not shown). The result suggests that there may be other regulation mechanism coupling Cln3 concentration to cell size. When the inducible promoter is repressed to different extent by reducing IPTG concentration, the correlation between Cln3 concentration and cell size is disrupted and promoter activity becomes the dominant controller of Cln3 concentration. In this work, we leave the question of how Cln3 is coupled to size and focus on how Start transition is triggered by a controllable Cln3 concentration.

Using Cln3* as the signal to trigger Start, we again observed a negative correlation between G1 length and Cln3 signal strength in single cells (*Figure 1E–H*): when the signal is strong, the cell passes Start sooner; when the signal is weak, the cell waits for a longer time. In *Figure 1I*, we plot G1 length vs the average Cln3 in G1 for many single cells. The data shows an inverse-like correlation. Similar results were obtained when mother–daughter cell types were distinguished, *CLN3D166A* was used as the signal, or G1 length is defined by the MCM marker (*Liku et al., 2005*) (*Figure 1—figure supplement 6–8*). All of these results strongly suggest that the inverse-like correlation between G1 length and Cln3 signal strength is an intrinsic property of the Start transition.

To get a quantitative understanding, we plot $\log(T_{G1} - T_0)$ against $\log(<Cln3> - Cln3_c)$ in *Figure 1J*, where $T_0$ is the horizontal asymptote representing the minimum G1 length (time from Whi5 nuclear entry to cytokinesis) and $Cln3_c$ is the vertical asymptote representing the minimum Cln3 concentration for cell to eventually pass Start. The slope of the linear fit is very close to −1 ($R^2 = 0.91$) (*Figure 1J* and

*Figure 1—figure supplement 9*). Thus G1 length is inversely proportional to the average Cln3 concentration in G1, which can be expressed as,

$$T_{G1} - T_0 = \frac{A}{<Cln3> - Cln3_c},$$ (1)

where *A* is the constant of proportionality. *Equation (1)* can be rewritten into an integral form:

$$\int_{T_0}^{T_{G1}} \left( Cln3(t) - Cln3_c \right) dt = A.$$ (2)

The above equation implies that the Start transition is triggered (at $t = T_{G1}$) when the time integration of the Start signal is above a threshold (*A*).

## Time integration reduces the variability in G1 length

To investigate the potential benefit of having signal integration during the cell cycle commitment, we simulated the mRNA expression and protein level of Cln3 with a stochastic algorithm (*Gillespie, 1976*). Most parameters in the model were derived from published papers (*Figure 2—source data 1*). Due to the low abundance and short half-life, the Cln3 profile fluctuates considerably (*Figure 2A*). We compared two hypothetical models of Start triggering: the Instantaneous Model in which the Start transition is triggered once the Cln3 level reaches a certain threshold, and the Integration Model in which the Start is triggered once the integration of Cln3 level over time reaches a certain threshold (*Figure 2A*). The Instantaneous Model leads to a large variability in G1 length even in identical cells (no extrinsic noise from cell-to-cell variability) and under a constant environment (*Figure 2B*). When extrinsic noise is added in the model, the G1 variability of the Instantaneous Model is as large as 92%, which is significantly larger than experimental observations for WT cells in previous and this studies (*Di Talia et al., 2007*; *Skotheim et al., 2008*; *Charvin et al., 2008*; *Ferrezuelo et al., 2012*) (CVs are no more than 50%, *Figure 2C*). In contrast, since the G1 length in the Integration Model depends on the integration of Cln3 level within a time window, a considerable amount of noise is averaged out. As a result, the variability in G1 length is much smaller (*Figure 2A–C*) and is consistent with the experimental data (*Figure 2C*). Furthermore, the shape of the G1 length distribution generated by the Integration Model is more similar to that of the experimental data. While the Instantaneous Model leads to a G1 length distribution with a long tail; this means that a significant fraction of the cells would have prolonged G1 lengths, which could be disadvantageous for the population. The conclusion holds the same when considering the nuclear volume increase during cell growth (*Figure 2—figure supplement 1*) and using a wide range of parameters.

We tested the Instantaneous Model with the measured Cln3 profiles. If the Instantaneous Model were true, cells should pass Start at or near the peak of Cln3 profile during G1. However, we found that in near 80% cells, the timing of Cln3 peak is different from the timing of Start (*Figure 2—figure supplement 2*). Thus, it is very unlikely that Start is triggered by the instantaneous Cln3 concentration.

## Whi5 acts as the integrator of Cln3-Cdk1 activity

Integration of Cln3 profile would imply an accumulation of Cln3-Cdk1 kinase activity over time. We next proceeded to identify the integrator, that is, on what molecule this kinase activity is being accumulated. The Start network is composed of two modules: Cln3-Cdk1 phosphorylating Whi5 as the triggering module and the positive feedback loop as the switching module (*Figure 1A*). The two modules are coupled via Whi5. The triggering module initiates Start by reducing Whi5 concentration in the nucleus. The switching module sets a threshold for the nuclear Whi5 concentration, $Whi5_c$, below which the switch will be flipped. Thus, G1 length is the time needed for the nuclear Whi5 concentration to drop below $Whi5_c$. We constructed a mathematical model for the kinetics of Whi5 phosphorylation (*Supplementary file 1A*). The model can reproduce the observed experimental data and make several predictions. The basic feature of the integration mechanism can be captured by a simplified version of the model as below.

In the simplified model, we assume there is only one phosphorylation site on Whi5, which is phosphorylated by Cln3-Cdk1 and dephosphorylated by some basal phosphatase. We further assume both enzymes operate at saturation (see *Supplementary file 1A* on how the assumption can be justified

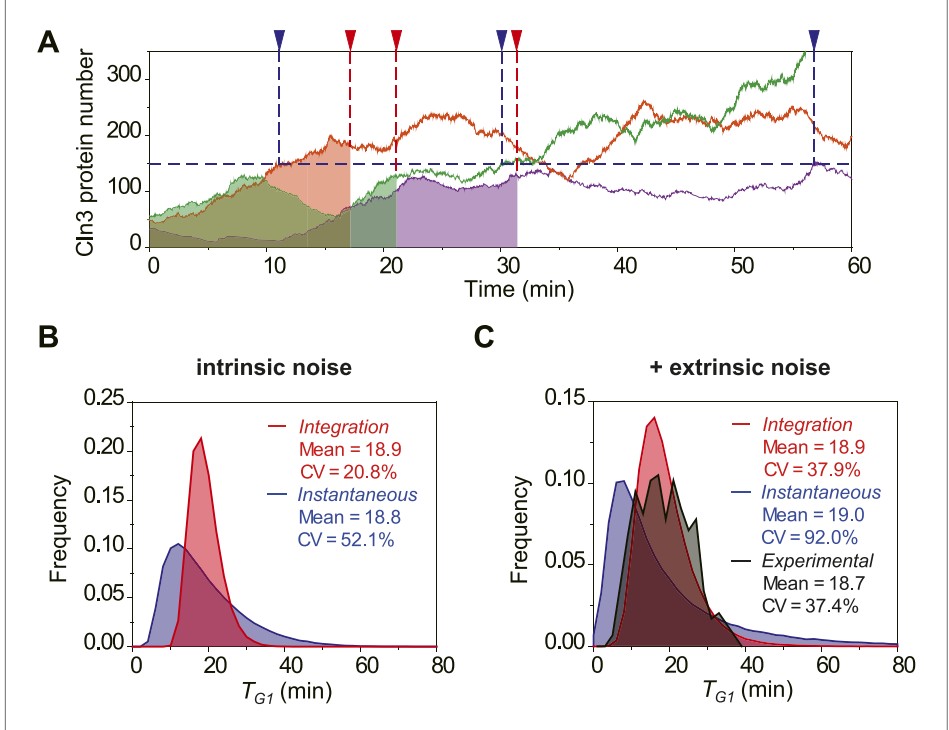

**Figure 2**. The integration of Cln3 reduces the variability of G1 length. (**A**) Representative Cln3 profiles in single cells. Each color represents simulation of a single cell. In the Instantaneous Model, Start is triggered at the time $T_{G1}$ (vertical blue dash line) when Cln3 profile hits a threshold (here set to 150) for the first time. In the Integration Model, Start is triggered at the time $T_{G1}$ (vertical red dash line) when the integration of Cln3 (the area under the Cln3 curve as indicated by shadow) reaches a threshold (here set to 1900). The two thresholds are chosen to generate the same average $T_{G1} \approx 20$ min in both models. In generating Cln3 profiles, both intrinsic noise and extrinsic noise are used (***Elowitz et al., 2002***; ***Raser and O'Shea, 2004***). (**B–C**) The distributions of $T_{G1}$ in the Instantaneous (blue) and Integration (red) models, respectively, with intrinsic molecular noise only (**B**) and with both intrinsic and extrinsic noise (**C**). The parameters to generate this figure are specified in ***Figure 2—source data 1***. The G1 length distribution from experimental data (Strain YCT2001) is shown in (**C**).

The following source data and figure supplements are available for figure 2:

**Source data 1**. Meaning, value and reference of the parameters to generate ***Figure 2***.

**Figure supplement 1**. Stochastic simulation of Cln3 profile and Start triggering process with nucleus volume increase.

**Figure supplement 2**. The Instantaneous Model fails with the Cln3 profiles measured in experiment.

**Figure supplement 3**. Measuring the memory length through Whi5 nuclear entry when Cdk1 is inactivated.

---

and/or relaxed). Then the rate change of the phosphorylated Whi5, $Whi5_P$, can be expressed by the following equation:

$$\frac{dWhi5_P}{dt} = k_1 \cdot Cln3 - k_2, \tag{3}$$

where $k_1$ is the catalytic rate of Cln3-Cdk1 on Whi5, and $k_2$ is the rate of the phosphatase times the phosphatase concentration. The concentrations of unphosphorylated and phosphorylated Whi5 fulfill a mass equation:

$$Whi5 + Whi5_P = Whi5_{tot}. \tag{4}$$

By our definition, G1 starts from Whi5 nuclear entry, when Cdc14 is the major phosphatase that is responsible for Whi5 dephosphorylation (*Taberner et al., 2009*). Cln3 cannot initiate the export of nuclear Whi5 until Cdc14 is inactivated at cytokinesis (*Shou et al., 1999*; *Visintin et al., 1999*). We define the time from Whi5 nuclear entry to Cdc14 inactivation as $T_0$. We assume $Whi5_P$ ($t = T_0$) = 0, thus $Whi5_{tot}$ is the nuclear Whi5 concentration when the phosphorylation reaction becomes dominant. At the Start transition $t = T_{G1}$, the nuclear Whi5 concentration drops to $Whi5_c$, which is the critical Whi5 threshold determined by the positive feedback loop (*Supplementary file 1A*), so that $Whi5_P$ ($t = T_{G1}$) = $Whi5_{tot} - Whi5_c$. By integrating *Equation (3)* from $T_0$ to $T_{G1}$, we have:

$$Whi5_{tot} - Whi5_c = \int_{T_0}^{T_{G1}} \left( k_1 Cln3 - k_2 \right) dt. \tag{5}$$

*Equation (5)* can be reformulated as:

$$\int_{T_0}^{T_{G1}} \left( Cln3 - Cln3_c \right) dt = A, \tag{6}$$

where $A = (Whi5_{tot} - Whi5_c)/k_1$ and $Cln3_c = k_2/k_1$. We thus identified Whi5 as the integrator: the number of phosphorylated Whi5, $Whi5_P$ accumulates with Cln3-Cdk1 activity and the increase of $Whi5_P$ causes a decrease in the nuclear Whi5 concentration. The Start transition happens when the nuclear Whi5 concentration drops below $Whi5_c$, or the accumulation of $Whi5_P$ reaches a threshold. In the case where the phosphatase operates in the linear region, *Equation (3)* becomes

$$\frac{dWhi5_P}{dt} = k_1 \cdot Cln3 - k_2 Whi5_P. \tag{7}$$

In this case, there is a window of 'memory' of length $1/k_2$, beyond which the integration effect is erased.

It is difficult to measure the window of memory or the memory length in G1 directly. Because most Whi5 is dephosphorylated and resides in the nucleus in G1 phase, we could not directly measure Whi5 dephosphorylation rate in G1. Thus, we measured Whi5 nuclear entry right after G1 by inhibiting Cdk1 activity with a strain bearing a *cdc28-as1* allele (*Bishop et al., 2000*). The average half time of Whi5 nuclear entry, which is an estimate of the memory length in the mathematical model (*Supplementary file 1A*), is 13.7 min in mother cells and 10.6 min in daughter cells, respectively (*Figure 2—figure supplement 3*). Note that this memory length is comparable to the average G1 length (from cytokinesis to Start) in daughter cells in SD medium, which is 13.6 min in our experiment. In poor nutrient conditions, when G1 length is prolonged, Whi5 dephosphorylation rate and the memory length might be further adjusted.

## The correlation between Cln3 and G1 length with perturbations in the Start network

The mathematical model of Whi5 kinetics makes several predictions. The first is that there is a linear relation between the integration threshold $A$ and the total Whi5 concentration $Whi5_{tot}$ (a more detailed ODE model predicts that $Cln3_c$ will also change with $Whi5_{tot}$ [*Supplementary file 1A*]). The second is that increasing the phosphatase activity $k_2$ reduces memory length and increases $Cln3_c$. The third is that reducing $Whi5_c$ by weakening the positive feedback loop increases the integration threshold $A$ (the ODE model predicts that $Cln3_c$ will also change when perturbing the loop strength [*Supplementary file 1A*]). Note that Cln3 half-life by itself has no effect on the integration dynamics; and that reducing the Cln3-Cdk1 catalytic efficiency $k_1$ (as in Cln3*) increases $Cln3_c$ and $A$ proportionally, requiring an increased Cln3 concentration to pass Start.

We verified the model's predictions by perturbing the Start network. First, the integration effect would have difficulty to manifest itself without the integrator. Indeed, our experiment shows that in *whi5Δ* strain, most cells pass Start with minimal G1 length (*Figure 3—figure supplement 1A*). We further checked to what extent the inverse correlation holds by plotting G1 length and Cln3 intensity in log–log scale (*Figure 3A*). The slope of the linear fit is much larger than −1, suggesting that the inverse proportionality is severely compromised. Second, the integration threshold $A$ should increase with the total Whi5 concentration. This means that at the same Cln3 concentration cell waits longer time in G1 with higher Whi5 dosage. The prediction is consistent with the previous finding that Whi5

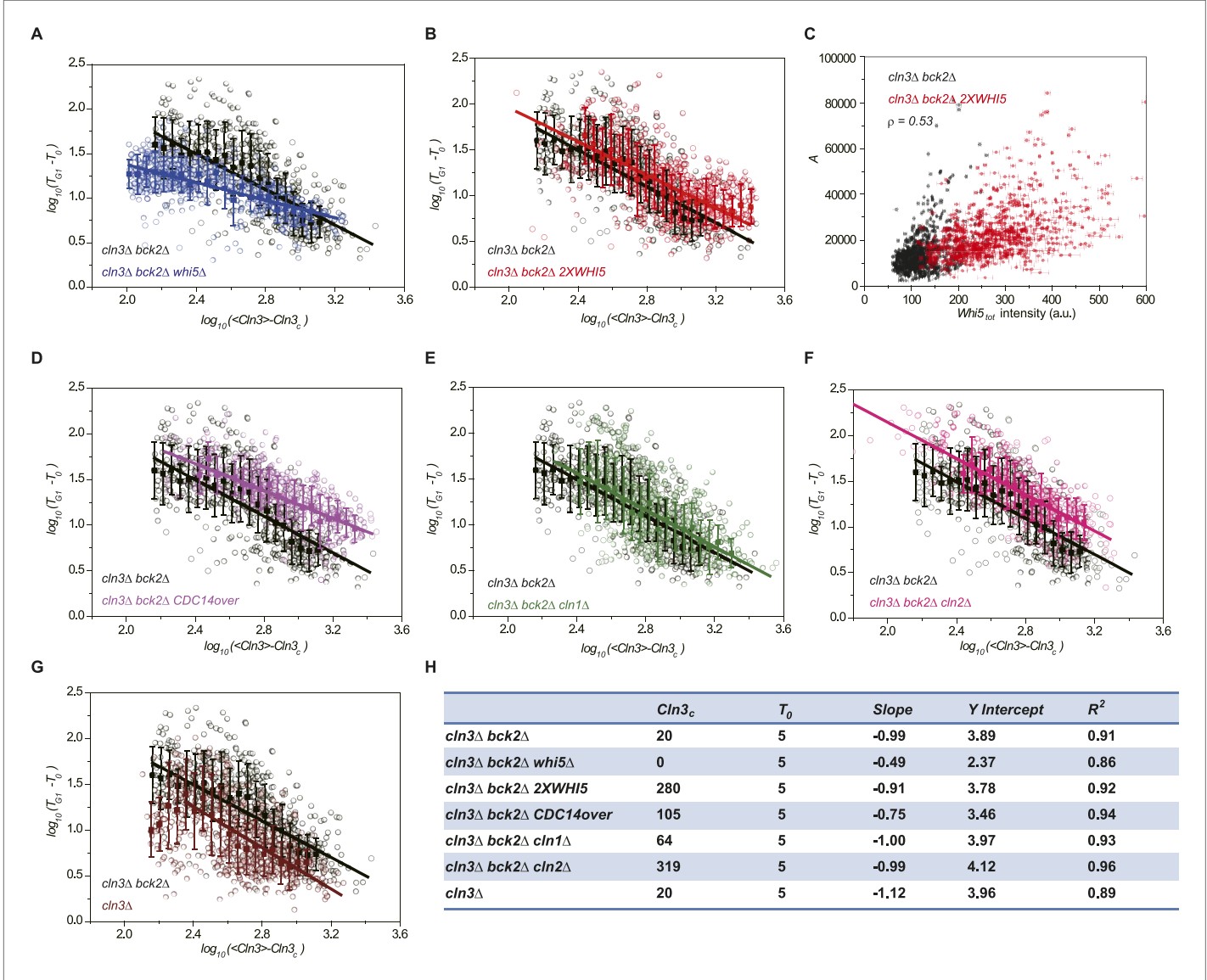

**Figure 3**. Whi5 acts as the integrator of Cln3-Cdk1 activity. Correlations between G1 length and the average Cln3 fluorescence in log–log scale in (**A**) *cln3Δ bck2Δ whi5Δ* strain (Strain YCT2008) (blue), (**B**) *cln3Δ bck2Δ 2XWHI5* strain (Strain YCT2007) (red), (**D**) *cln3Δ bck2Δ CDC14* overexpressing strain (Strain YCT2009) (purple), (**E**) *cln3Δ bck2Δ cln1Δ* strain (Strain YCT2013) (green), (**F**) *cln3Δ bck2Δ cln2Δ* strain (Strain YCT2014) (pink) and (**G**) *cln3Δ BCK2+* strain (YCT2015) (wine), in comparison with in *cln3Δ bck2Δ* strain (Strain YCT2003) (black). Data are also plotted in linear scale in *Figure 3—figure supplement 1*. <*Cln3*> represents the average Cln3 fluorescence in G1; $T_0$ and $Cln3_c$ are fitted from *Figure 3—figure supplement 1* as described in 'Materials and methods'. The solid lines are linear fits of the binned data (filled squares). Error bars indicate standard deviation. $Cln3_c$, $T_0$, Slope, Y Intercept and $R^2$ of the linear fit for each strain are summarized in (**H**). (**C**) The integral A (calculated as $T_{G1}$ mutiplied by average Cln3 intensity in G1) vs $Whi5_{tot}$ fluorescence intensity in *cln3Δ bck2Δ* (black) and *cln3Δ bck2Δ 2XWHI5* (red) strains. Each dot represents a measurement from one cell cycle event; error bars indicate standard deviation ρ signifies Pearson correlation coefficient.

The following figure supplements are available for figure 3:

**Figure supplement 1**. Correlations between G1 length and the average Cln3 fluorescence in (**A**) *cln3Δ bck2Δ whi5Δ* strain (Strain YCT2008) (blue), (**B**) *cln3Δ bck2Δ 2XWHI5* strain (Strain YCT2007) (red), (**C**) *cln3Δ bck2Δ CDC14* overexpressing strain (Strain YCT2009) (purple), (**D**) *cln3Δ bck2Δ cln1Δ* strain (Strain YCT2013) (green), (**E**) *cln3Δ bck2Δ cln2Δ* (Strain YCT2014) (pink) and (**F**) *cln3Δ BCK2+* strain (YCT2015) (wine), in comparison with in *cln3Δ bck2Δ* strain (Strain YCT2003) (black).

**Figure supplement 2**. Variation of G1 length in low Cln3 region is due to the variance in $Whi5_{tot}$.

overexpression significantly increased the percentage of the cells in G1 phase (*Costanzo et al., 2004*). We quantitatively verified the effect of the total Whi5 concentration on the integration threshold $A$ with a strain containing two copies of *WHI5* (*Figure 3—figure supplement 1B* and *Figure 3B*). The inverse correlation shifts upward in log–log scale, which represents larger $A$. Remarkably, the positive correlation between $A$ and Whi5 concentration can also be seen at the single-cell level (Pearson correlation coefficient = 0.53, *Figure 3C*). In Cln3-$T_{G1}$ correlations, the variation of G1 is high when Cln3 concentration is low. At least part of the variation is due to the variance in $Whi5_{tot}$ concentration (and thus the variance in $A$). After normalizing by $Whi5_{tot}$, inverse correlations in both *1X* and *2X WHI5* strains are more converged and collapse to the same curve (*Figure 3—figure supplement 2*). Third, the model predicts that increasing the phosphatase activity will not only increase $Cln3_c$ but also cause deviation from the inverse correlation by shortening the memory length. It is known that overexpressing the phosphatase *CDC14* affects the nuclear accumulation of Whi5 and increases the percentage of the cells in G1 phase (*Visintin et al., 1998*; *Stevenson et al., 2001*). Thus, we tested the prediction in a *CDC14* overexpressing strain. We observed that $Cln3_c$ increases and the inverse proportionality is compromised (*Figure 3—figure supplement 1C* and *Figure 3D*). Finally, the model predicts that weakening the strength of the positive feedback loop will both reduce $Whi5_c$ (thus increase $A$) and increase $Cln3_c$. We measured G1 length vs Cln3 intensity in *cln1Δ* and *cln2Δ* strains and observed that $Cln3_c$s increase and the inverse correlations shift upward in log–log scale as predicted (*Figure 3—figure supplement 1D,E* and *Figure 3E,F*). The correlation in *cln2Δ* shifts more than in *cln1Δ*, suggesting that Cln2 contributes more to the positive feedback loop. This is consistent with the fact that the expression level of Cln2 is about three times higher than Cln1 and with the previous finding that Cln2 is more potent in the Start transition (*Huh et al., 2003*; *Tyers and Futcher, 1993*; *Tyers et al., 1993*).

We also investigated how the correlation between Cln3 and G1 length is affected by Bck2, which triggers the Start transition in the absence of Cln3 (*Wijnen, 1999*). As shown in *Figure 3—figure supplement 1F*, the Cln3-$T_{G1}$ correlation of *BCK2+* strain loses the long G1 tail at low Cln3 concentration. And in log–log scale the inverse correlation only holds to certain range. As Cln3 concentration decreases, the mean G1 length stops increasing at about 29 min. The result suggests that in wild-type cells, Bck2 acts as a bypass Start trigger to prevent a too long G1 phase when Cln3 concentration is too low.

## Cells modulate Whi5 concentration in different nutrient conditions to coordinate G1 length with growth rate

In budding yeast, the G1 checkpoint Start coordinates cell division with growth (*Hartwell et al., 1974*; *Jorgensen and Tyers, 2004*). Under different nutrient conditions, cells spend different time growing in G1 before passing Start (*Ferrezuelo et al., 2012*). However, it is not clear how this coordination is achieved. In light of our findings (*Equation (5)*, *Figure 3B,C*), it is suggestive that the cell could modulate the integration threshold $A$, and thus the G1 length, by tuning $Whi5_{tot}$. Indeed, we found that in suboptimal nutrient conditions, the Cln3-$T_{G1}$ correlations shift towards right with higher $Whi5_{tot}$ intensity and $A$ (*Figure 4—figure supplement 1*). Similar shift was observed with the MCM marker as a measure of G1 length (*Figure 4—figure supplement 3*). However, we did not observe any increase of MCM intensity in suboptimal nutrient condition, nor any correlation between $A$ and MCM intensity, suggesting that the nutrient regulation on Whi5 and the Whi5 regulation on $A$ are specific. We further quantified $Whi5_{tot}$ intensity in a series of nutrient conditions in WT cells. Surprisingly, $Whi5_{tot}$ increases as growth rate decreases and G1 length is accordingly prolonged in both mother and daughter cells (*Figure 4A–C*). It means that in poor nutrient conditions, the Start threshold is actually higher rather than lower as proposed by previous studies (*Jorgensen and Tyers, 2004*; *Schneider et al., 2004*; *Turner et al., 2012*), although the cell size may appear to be smaller.

We also monitored the adaptation of $Whi5_{tot}$ intensity and G1 length after nutrient change. When the cells were switched from low glucose to high glucose medium, both $Whi5_{tot}$ intensity and G1 length decreased gradually and reached the same level as in constant high glucose medium after several cell cycles (*Figure 4—figure supplement 6A–C,F,G*). Contrarily, when cells were switched from high glucose to low glucose medium, $Whi5_{tot}$ increased fast and underwent an overshoot before it reached a steady level (*Figure 4—figure supplement 6D,H*). G1 length increased even faster and its overshoot was also more pronounced than Whi5 (*Figure 4—figure supplement 6D,I*). Surprisingly, we found when switched from high to low glucose, if the cell had not started budding, G1 phase could reinitiate (*Figure 4—figure supplement 6E*). This result could not be explained by current knowledge of Start and must involve further changes in CDK and phosphatase activities.

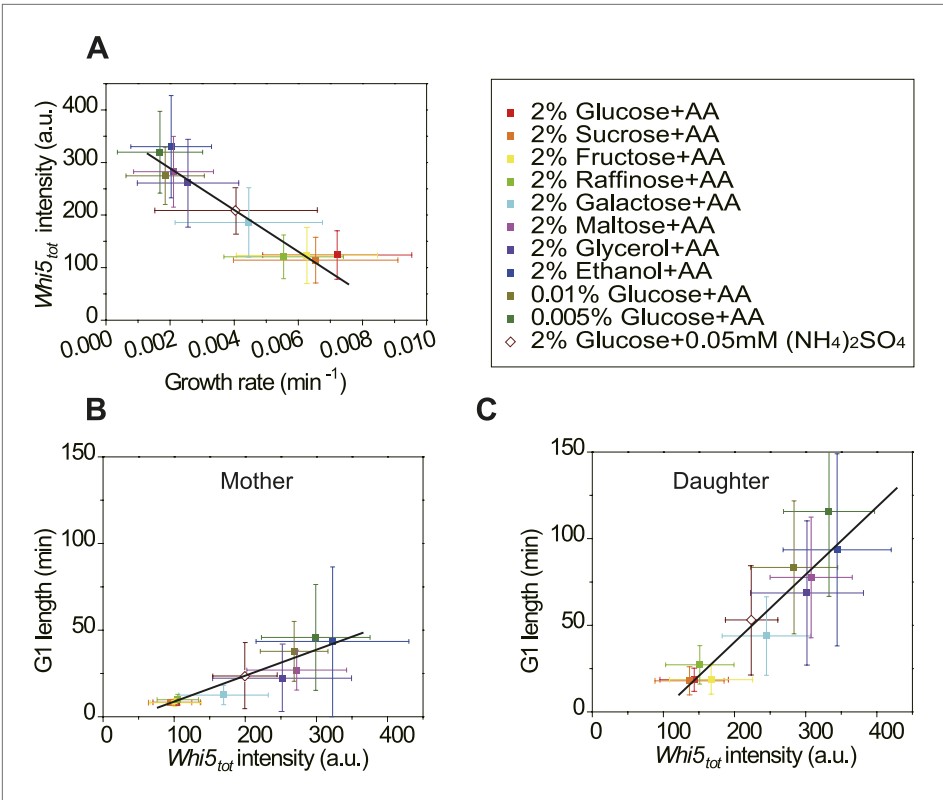

**Figure 4**. Cells modulate $Whi5_{tot}$ concentration to coordinate cell division with nutrient conditions. (**A**) $Whi5_{tot}$ intensity is negatively correlated with growth rate in various nutrient conditions (Strain YCT2001). (**B–C**) G1 length is positively correlated with $Whi5_{tot}$ intensity for mothers (**B**) and daughters (**C**) (Strain YCT2001) in different nutrient conditions. Each color represents one nutrient condition as in **A**. Each dot in **A–C** is calculated from all cells in that nutrient condition. (Single-cell data are presented in *Figure 4—figure supplement 4,5*; statistic results are summarized in *Figure 4—source data 1*) Black straight lines are guide lines; error bars indicate standard deviation.

The following source data and figure supplements are available for figure 4:

**Source data 1**. (**A**)The growth rates and $Whi5_{tot}$ intensities in different nutrient conditions.

**Figure supplement 1**. The correlations between Cln3 concentration, $Whi5_{tot}$ intensity and G1 length in suboptimal nutrient conditions.

**Figure supplement 2**. The correlations between the integral A and $Whi5_{tot}$ fluorescence intensity in 2% glucose (**A**), 2% raffinose (**B**), limited nitrogen (**C**) and 0.05% glucose (**D**) (Strain YCT2003).

**Figure supplement 3**. The inverse correlations in raffinose comparing with in glucose with the MCM marker as a measure of G1 length (YCT2010).

**Figure supplement 4**. The correlation between growth rate and $Whi5_{tot}$ fluorescence intensity in single cells in different nutrient conditions (YCT2001).

**Figure supplement 5**. The correlation between $Whi5_{tot}$ intensity and G1 length in single cells in different nutrient conditions in mother (**A**) and daughter (**B**) cells (YCT2001).

**Figure supplement 6**. The adaptation of Whi5 and G1 length after nutrient transitions (YCT2001).

## Discussion

### Time integration in decision making process

Cellular decision making is a fundamental problem with broad implications, ranging from development, cell-fate determination, stress response, to cell cycle control and signaling. A decision-making process in a cell usually consists of (i) sensing (of external and/or internal signals and cues), (ii) information processing (of the sensed information), and (iii) actuating (e.g., turning on a switch). Previous works have been mainly focused on the first and the last steps. Even in many cases where the molecular players and the circuitry are mapped out, how cells process information to make the best decision is largely unknown. This is intrinsically a quantitative question and is especially important when the signals are noisy and with uncertainties. Information processing is the brain of decision making; understanding this step is critical to understand the rationale and the strategy the cell adopts to make better decisions.

We address this question with a well-studied model system: the budding yeast Start checkpoint. Using a quantitative single cell assay with a controllable and quantifiable Cln3 signal, we discovered that it is not the instantaneous value of the Cln3 concentration, but rather the integration of the concentration over time, that triggers the switch. Cln3 is a sensor that senses, with a fast response time, the instant information/condition relevant to making cell cycle commitment. We found that the instant Cln3 activity is memorized on phosphorylated Whi5, and the memory length is about 10 min in SD medium. This implies that the cell uses information within a time window of the past in order to get an assessment of the future.

This strategy of signal integration averages out noise and fluctuations and minimizes error and uncertainty in decision making. Signal integration has been seen in decision-making behaviors of animals (*Bowman et al., 2012*; *Bogacz et al., 2006*). Our work shows that it is also adopted at the cellular level, suggesting a general strategy that may be widely implemented in decision-making and signaling systems (*Li et al., 2010*; *Di Talia and Wieschaus, 2012*; *Doncic and Skotheim, 2013*).

### Novel roles of Whi5 in the Start transition

Whi5 (more precisely, the phosphorylated Whi5) serves as the integrator in the system, that is, the physical memory onto which the integration is being recorded. We found that cells modulate Whi5 level to set different integration thresholds in different nutrient conditions. Whi5 level is higher in poor nutrients, thus cells are more cautious and wait for longer time to make the decision.

We also found that daughter cells always have higher $Whi5_{tot}$ intensity and steeper slope between $Whi5_{tot}$ and G1 length than mother cells (*Figure 4D,E*). The finding suggests that besides the daughter-specific Cln3 suppression by Ace2 (*Laabs et al., 2003*), Whi5 contributes to G1 delay in daughter cells as well. It is also consistent with the previous finding that Whi5 is required for the different behaviors of mother and daughter cells under low metal stress (*Avraham et al., 2013*).

Although Cln3 and Whi5 determine G1 length together, they have very different response time. While Cln3 responds to external and internal cues rapidly, Whi5 adjusts in a much longer time scale and reinforces the change of G1 length (*Figure 4—figure supplement 6*). The system could utilize the different response times of the sensor and the integrator to develop more complex strategies when adapting to environmental fluctuations and changes.

### Revisiting the size control problem

The coordination of cell growth with division is often posed as a 'size control problem'. Previous studies on size control were more focused on Cln3's response to environmental changes (*Gallego et al., 1997*; *Polymenis and Schmidt, 1997*; *Hall et al., 1998*; *Parviz et al., 1998*; *Jorgensen and Tyers, 2004*). It was proposed that the Start threshold is lowered in poor nutrient conditions, based on the conventional (instantaneous) model and the observation that the abundance of Cln3 is lower in these conditions (*Jorgensen and Tyers, 2004*; *Schneider et al., 2004*; *Turner et al., 2012*). However, our finding on Whi5 modulation reveals a different scenario – the Start threshold is in fact raised in poor nutrients and the longer G1 length is a result of a longer integration time of Cln3. Cell cycle is coupled to growth rate by both Cln3 and Whi5. A full understanding of size control may need to take into account the combined action of Cln3 dynamics, $Whi5_{tot}$ concentration and the integration mechanism.

## Materials and methods

### Strains and plasmids

Standard methods were used throughout.

All strains in this study are congenic *W303*. *W303-1A ADE2+* was made by integrating *ADE2* fragment amplified from the *4741* genome at the *ADE2* locus of *W303-1A*. The *KAN-MX6*, *NAT-MX6*, and *LEU2* fragments flanking with homologous sequence to the target gene (40 bp) for deletion were amplified from the plasmid pFA6-KAN-MX6 (*Longtine et al., 1998*), pFA6-NAT-MX6 (*Goldstein and McCusker, 1999*), and pRS305 (*Sikorski and Hieter, 1989*), respectively. The *WHI5-tdTomato* constructs were made by digesting pCT2001 with HindIII, integrating at the *WHI5* locus and losing the *CaURA3* marker between the two *TEF1* terminators. The inducible *CLN3* constructs were made by digesting pCT2002, pCT2003, pCT2004, or pCT2005 with PmeI and integrating at the *HIS3* locus. The *ADH1pr-HTB2-CFP* construct was made by digesting pCT2006 with XbaI and integrating at the *TRP1* locus. The *ADH1pr-MCM-mCherry*, *ADH1pr-MCM-GFP*, additional *WHI5pr-WHI5-tdTomato* and *GPDpr-CDC14* constructs were made by digesting pCT2007 with NdeI, pCT2008 with BstBI, pCT2009 with NdeI and pCT2010 with NcoI, respectively and integrating at the *URA3* locus. All the constructs and deletion strains were verified by PCR. All the strains used in this study are summarized in *Supplementary file 2A*.

The plasmid pCT2001 was constructed as following: first replaced the *ADH1ter* between AscI and BglII on the plasmid pNI8, a kind gift from Jonathan Weissman (UCSF), with the *TEF1ter* amplified from the same plasmid; then replaced the *mCherry* fragment between PacI and AscI with the *tdTomato* fragment amplified from pRS304-tdTomato; next the 300 bp upstream of the *WHI5* stop codon amplified from the *W303* genome was inserted between HindIII and PacI sites; finally the 300 bp downstream of the *WHI5* stop codon amplified from the *W303* genome was inserted by using ClaI, along with an additional HindIII site at the 3′ end. Each inducible Cln3 plasmid contains three transcription units: the lactose transporter, the constantly expressed LacI and the inducible promoter controlled Cln3 signal. The *STE5pr* (−602 to −1 of *STE5*) amplified from the *W303* genome, the *LAC12* amplified from pKR1B-LAC4-1 (*Sreekrishna and Dickson, 1985*), and the *CYC1ter* amplified from pGREG506 (*Jansen et al., 2005*) were inserted into pRS304 (*Sikorski and Hieter, 1989*) with SacI-SpeI-SalI-XhoI to make the transcription unit of the transporter (each fragment was sequentially inserted between two restriction sites). The *GPDpr* amplified from pRS424-GPD (*Mumberg et al., 1995*), the *LacI* gene amplified from the *Escherichia coli* genome and the *CYC1ter* were inserted into pGREG506 with NotI-HindIII-EcoRI-SacI to make the transcription unit of LacI. The inducible promoter *GlacSpr*, which essentially is the *ADH1pr* (−700 to −1 of *ADH1*) with two LacI binding sites (one on each side of the TATA box), was obtained by overlapping PCR (*Figure 1—figure supplement 2*). *GFP* was amplified from pNT10, a kind gift from Jonathan Weissman (UCSF). *Venus* was amplified from pVenus-N1-NPY (*Nagai et al., 2002*). A 11-amino-acid linker optimized for yeast (Ala-Ala-Ala-Gly-Asp-Gly-Ala-Gly-Leu-Ile-Asn-) was introduced to the C terminal of the fluorescent proteins by PCR primers. The wild-type *CLN3* was amplified from the genome, while mutants were constructed by overlapping PCR. The transcription unit of Cln3 signal *GlacSpr-CLN3-CYC1ter* was constructed by inserting the corresponding fragments sequentially into pGREG506 with ClaI-BamHI-SalI-XhoI; *GlacSpr-GFP-GFP-CLN3R108A-CYC1ter* and *GlacSpr-GFP-GFP-CLN3D166A -CYC1ter* were constructed into pGREG506 with ClaI-BamHI-EcoRI-NotI-SalI-XhoI; *GlacSpr-Venus-Venus-Venus-CLN3R108A-CYC1ter* was constructed into pGREG506 with ClaI-BamHI-EcoRI-SphI-NotI-SalI-XhoI. Then, the transcription units of LacI and Cln3 were subcloned into pNH603 (a kind gift from Wendell A. Lim lab, designed to make sure single integrant) with SacI-NotII and ClaI-XhoI respectively. Finally, the transcription unit of transporter was amplified by PCR and inserted into the plasmid with SacI site to accomplish the inducible Cln3 plasmids pCT2002-pCT2005. The pCT2006 plasmid was constructed by inserting the *ADH1pr* amplified from the genome, the *HTB2* without stop codon amplified from the genome, the *CFP* with linker at the N-terminal amplified from BBa_E0020 (iGEM Registry) and the *CYC1ter* into pRS304 with SacI-NotI-SpeI-SalI-XhoI. The pCT2007 and pCT2008 plasmids were constructed by inserting the *ADH1pr*, the *MCM* marker amplified from pML103 (*Liku et al., 2005*), *GFP* (for pCT2007), or *mCherry* (for pCT2008) with linker at the N terminal and the *CYC1ter* into pRS306 (*Sikorski and Hieter, 1989*) with SacI-NotI-NotI-SalI-KpnI. The pCT2009 plasmid was constructed by replacing the *mCherry* fragment between PacI and AscI on pNT8 with *tdTomato*, inserting the upstream 1430 bp of the *WHI5* stop codon (including *WHI5pr* and *WHI5* coding sequence) amplified from the genome into HindIII-PacI and subcloning the HindIII-BglII fragment

of the resultant plasmid into pRS306. pCT2010 was constructed by inserting the *GPDpr*, the *CDC14* amplified from the genome and the *CYC1ter* into pRS306 with SacI-BamHI-SalI-XhoI. All the plasmids were verified by sequencing on both strands and summarized in *Supplementary file 2B*.

## Media and chemicals

All the inverse correlations were measured in Synthetic Dextrose (SD) medium containing 2% (wt/vol) glucose, 1× amino acid (AA) dropout, 6.7 g/L yeast nitrogen base (YNB) without amino acid, 100 mg/L leucine, 20 mg/L histidine, 20 mg/L tryptophan, 20 mg/L adenine, and 20 mg/L urea, unless otherwise indicated. The formula of 1× amino acid dropout is 20 mg/L arginine, 20 mg/L methionine, 30 mg/L tyrosine, 30 mg/L isoleucine, 30 mg/L lysine, 50 mg/L phenylalanine, 100 mg/L glutamic acid, 100 mg/L aspartic acid, 150 mg/L valine, 2 g/L threonine and 4 g/L serine. The limited nitrogen medium contains 2% (wt/vol) glucose, 0.05 mM ammonium sulfate, 6.7 g/L yeast nitrogen base (YNB) without ammonium sulfate and amino acid, 100 mg/L leucine, 20 mg/L histidine, 20 mg/L tryptophan, 20 mg/L adenine, and 20 mg/L urea (*Gallego et al., 1997*). The limited carbon media have similar ingredients as the SD medium except for various carbon sources or glucose concentrations. All nutrients were purchased from Sigma–Aldrich, St. Louis, MO, except that glucose was purchased from Ameresco, Solon, OH. IPTG (Isopropyl β-D-1-thiogalactopyranoside) was purchased from Sigma–Aldrich, St. Louis, MO and dissolved to make 0.5 M stocks. conA was purchased from Sigma–Aldrich, St. Louis, MO and dissolved to make 1 mg/mL stocks. 1-NM-PP-1 was purchased from Merck Millipore, Billerica, MA and dissolved in DMSO to make 5 mM stocks.

## Constructing the homologous structure of Cln3-Cdk1 complex and screening for desired Cln3 mutants

We constructed the homologous structure of Cln3 based on the cyclin A chain (chain B) in 3DDQ (PDB ID). Two other structures, the cyclin D3 chain (chain B) in 3G33 (PDB ID) and the cyclin E1 chain (chain B) in 1W98 (PDB ID), were chosen for reference as well. The amino acid sequences of those proteins were aligned by the 3D-Jury server (http://meta.bioinfo.pl/) (*Ginalski et al., 2003*), and then the alignment result was used to model the 3D structure of Cln3 in Rosetta CM (comparative modeling, v37268) (*Chivian and Baker, 2006*). Similar method was used to construct the homologous structure of Cdk1, with the Cdk2 chain (chain A) in 1VYW (PDB ID) as the template. The complex model of Cln3 and Cdk1 was prepared by Rosetta Docking (*Wang et al., 2007*), taking the complex structure of Cdk2-cyclin A (PDB ID: 1FIN) as the template. The contacts or residues in the complex models were optimized by Rosetta Relax with backbones fixed. The importance of the interface residues was evaluated by using the fixed-backbone alanine scanning protocol in Rosetta (*Kortemme and Baker, 2002*).

To screen for the desired Cln3 mutants, we selected 12 single amino acid sites in clustered charged residues (*Miller et al., 2005*). Three more sites were chosen from the conserved MRAIL hydrophobic patch involved in substrate reorganization (*Schulman et al., 1998*). Five more mutation sites on the interface of Cln3-Cdk1 complex were suggested by in silico alanine scanning. Finally, we selected 20 single-amino-acid sites in total for alanine scanning experimentally. Mutants were constructed by site-directed mutagenesis PCR. The mutation sites were summarized and labeled on the homologous structure of Cln3 in *Figure 1—figure supplement 3*.

To check the brightness and the activity of the Cln3 mutants, the wild-type and mutant Cln3s were fused with two tandem GFPs on the N terminus and expressed by the synthetic inducible promoter *GlacSpr*. Each construct was integrated at the *HIS3* locus of the *cln3Δ* strain YCT2011. The GFP brightness and cell size under full induction was quantified by fluorescent microscopy. Only mutants with significantly higher GFP intensity than wild-type were shown in *Figure 1—figure supplement 4*. We further deleted *BCK2* in those strains and investigate their cell division without IPTG under microscopy. The shut-off of cell cycle was considered tight if less than 5% cells budded in 6 hr.

## Microfluidic device

Similar molds and methods as described previously (*Tian et al., 2012*) were used to fabricate the microfluidic chips. Medium was fed through the main channel by auto-controlled syringe pump (TS-1B, Longer Pump Corp., Baoding, China). The flow rate was 66.6 µL/hr. In the nutrient switching experiments, both the tubing and the syringe providing the medium were changed within 3 min, to make sure the transition is as fast as possible.

## Time lapse microscopy

Except for the measurement of Whi5 dephosphorylation rate, cells were grown in the microfluidic chip to maintain the constant IPTG concentration and nutrient condition. Temperature of the microfluidic chip was kept at 30°C with a stage top incubator (INU-TIZHB-F1, Tokai Hit Co., Ltd., Fujinomiya-shi, Japan).Time lapse movies were collected with epi-fluorescence microscopy using a Nikon Ti-E inverted microscope equipped with the objective lens Plan Apo VC 100×/1.40 Oil DIC N2, the motorized XY stage and the Perfect-Focus System (Nikon Co., Tokyo, Japan). Images were acquired every 3 min with an Andor iXon3 897 EMCCD (512 × 512, 16 μm, Andor Technology Ltd., Belfast, UK) and Lambda SC shutter controllers (Sutter Instrument, Novato, CA). NIS Elements AR v3.2 (Nikon Co., Tokyo, Japan) was used to automate image acquisition and microscope control. There is no significant photo-toxicity or perturbations of cell cycle time caused by the detection of either GFP-GFP-Cln3 or Whi5-tdTomato or both. There is no significant photo-bleaching in the GFP or tdTomato channel (data not shown). The brightness of the mercury lamp on different days was normalized by fluorescent reference slides (Fluor-REF, Microscopy Education, Microscopy & Imaging Place Inc., McKinney, TX).

To measure Whi5 dephosphorylation rate, cells were immobilized on glass bottomed dish by conA. Temperature was kept by ZILCS incubation chamber (Tokai Hit Co., Ltd., Fujinomiya-shi, Japan). Cells were monitored for 30 min and then another 2 hr after adding 50 μM 1-NM-PP-1 to the medium. Time lapse movie was collected with a UltraVIEW VoX Laser Confocal Imaging System (PerkinElmer, Watham, MA) and a CSU-X1 spinning disk confocal (Yokogawa,Tokyo, Japan) on a Nikon Ti-E inverted microscope equipped with the APO TIRF 100X OIL NA 1.45 objective lens, the motorized XY stage and the Perfect-Focus System (Nikon Co., Tokyo, Japan). Whi5-tdTomato fluorescence was excited with the 561 nm 50 mW laser line and collected by the appropriate filters. Images were acquired every 3 min by a Hamamatsu C9100-13 EMCCD (Hamamasu Photonics K. K., Hamamatsu City, Japan) camera. At each time point, 5 Z-series optical sections were collected with a step size of 1.6 μm, using a NanoScanZ 400 μm Piezo focusing drive (Prior Scientific, Cambrige, UK). Maximum projections of Z stacks were performed with ImageJ.

## Image and data analysis

Cell segmentation and tracing were based on bright field images and automatically accomplished by the MATLAB custom software *cellseg* as described previously (*Lau, 2009*; *Yang et al., 2013*). Daughters were counted in once the bud can be recognized by the software (usually in early or middle mitosis). We quantified the mean intensity of the brightest 5 × 5 Whi5-tdTomato pixels in one cell as nuclear Whi5 concentration. G1 length was defined as the time interval between the local maximum and minimum of its first derivative within certain time frame (as shown in *Figure 1—figure supplement 1*). In strains bearing the MCM marker, G1 length was defined similarly except for using the MCM marker. The MCM marker is an artificial marker generally reporting CDK activity. It enters the nucleus 2 min prior to and exports 5 min later than Whi5 (data not shown). $Whi5_{tot}$ intensity was taken as the maximum of nuclear Whi5 intensity during one G1 phase. Since Cln3 localizes in the nucleus, similar to Whi5, we quantified the mean intensity of the brightest 9 × 9 GFP-GFP-Cln3* pixels in one cell as Cln3 concentration, which had been verified to be a good proxy (data not shown). The brightest 3 × 3 pixels were deducted from the brightest 9 × 9 pixels to eliminate the artifact of aggregates. Cell size was taken as cell area in two dimensional. Cells were assumed grow exponentially in G1. Growth rate fitted by more than six data points and with standard error smaller than 0.02 were used in the final statistics. Export of the fluorescent intensities and cell area was automatically accomplished by *cellseg* with minor revision. Following analysis was accomplished by the MATLAB custom software *analyzestart*, which was developed before (*Lau, 2009*; *Yang et al., 2013*) and modified for the purpose of this study. The integral $A$ in *Figures 3C, 4B*, *Figure 4—figure supplement 1D,G,I, 2, 3B* was calculated as Cln3 intensity multiplied by G1 length. Whi5 dephosphorylation rate was fitted by the equation $Whi5(t) = A - B \cdot exp(-t/\tau)$, where $1/\tau$ is the Whi5 dephosphorylation rate and $\tau$ is the memory length; $A$ denotes $Whi5_{tot}$ concentration; $B$ denotes $Whi5_{tot} - Whi5\ (t = 0)$ (*Supplementary file 1A*).

## Fitting the slope of Cln3-$T_{G1}$ correlations in log–log scale

Cln3-$T_{G1}$ correlations in log–log scale were fitted by the following criteria:

1. $T_0$ was set as 5 min for strains with the Whi5 marker (*Di Talia et al., 2007*) or 12 min for strains with the MCM marker.

2. Except for *whi5Δ* and *BCK2+* strains, $Cln3_c$ was fitted in the range of *min_Cln3*-200 to *min_Cln3*. For each strain, *min_Cln3* is the lowest average Cln3 intensity we observed that can pass Start.
3. In *whi5Δ* strain, $Cln3_c$ was set as 0.
4. In *cln3Δ BCK2+* strain, $Cln3_c$ was set as the same value as *cln3Δ bck2Δ* strain.
5. Data points in log–log scale were binned according to their Cln3 intensity.
6. Only bins with more than 10 data points were used in fitting.
7. Median of Cln3 intensity and G1 length within each bin was used in fitting to avoid the effect of outliers.

For *cln3Δ bck2Δ* strain in 2% glucose, we adopted the least square fit. For other strains or growth conditions, we adopted the fit whose slope is closest to −1. For *cln3Δ BCK2+* strain, there was no good linear fit for the whole Cln3 intensity region, thus we only fitted the high Cln3 part.

## Stochastic simulation of the Cln3 profile and simple models of the Start triggering process

Stochastic simulation of the Cln3 profile was done using Stochastic Simulation Algorithm (SSA) converted from the Ordinary Differential Equations (*Gillespie, 1976*):

$$\frac{d[mRNA_{cln3}]}{dt} = a_1 - D_1 \cdot [mRNA_{cln3}]$$

$$\frac{d[Protein_{Cln3}]}{dt} = a_2 \cdot \exp(\alpha \cdot t) \cdot [mRNA_{cln3}] - D_2 \cdot [Protein_{Cln3}]$$

$$Initial\ state: [mRNA_{cln3}] = Int\left(\frac{a_1}{D_1} \cdot r\right), [Protein_{Cln3}] = Int\left(\frac{a_1 \cdot a_2}{D_1 \cdot D_2} \cdot r\right)$$

$$Int(x) = \{n \mid n \in \mathbb{Z}, \mid x - n \mid \leq 0.5$$

The meaning, value and reference of the parameters to generate *Figure 2* are summarized in *Figure 2—source data 1*.

We adopted flowing assumptions in generating Cln3 profile:

1. Constant transcription rate of CLN3 gene.
2. Constant degradation rate of Cln3 mRNA and protein.
3. Exponential growth of cell volume.
4. Ribosome number (reflected by translation rate) is proportional to cell volume.
5. Cln3 transcription gets suppressed at early G1 (*r* ratio of full capacity) to mimic the Ace2 suppression in daughter cells.
6. Nuclear volume is constant thus Cln3 abundance directly reflects Cln3 nuclear concentration. In the model, we simply simulated Cln3 mRNA and protein numbers instead of concentrations.

In Instantaneous Model, G1 starts from zero time point with Cln3's initial state. $T_{G1}$ is the time when $Protein_{Cln3}$ first exceeds *Instantaneous_threshold*. In Integration Model G1 starts from zero time point with Cln3 initial state. $T_{G1}$ is the time when the integration of Cln3 ($\int Protein_{Cln3} dt$) surpasses *Integration_threshold*. *Instantaneous_threshold* was chosen as 150, while *Integration_threshold* was chosen as 1900 to generate average 19 min $T_{G1}$.

Extrinsic noise was simulated by randomizing the model parameters around their nominal values within a certain percentage range. Extrinsic noise of 20% CV was applied to all parameters ($D_1$, $a_1$, $D_2$, $a_2$, $\alpha$, $r$, *Instantaneous_threshold* and *Integration_threshold*) within this model.

For the robustness of our conclusion, we checked our model with different parameter sets of {$D1$, $a_1$, $D_2$, $a_2$, $\alpha$} varying in all possible ranges and found that the Integration Model does generate a $T_{G1}$ distribution with much closer resemblance to experimental result than the Instantaneous Model under all circumstances.

We also considered the case that nuclear volume increases with cell volume in *Figure 2—figure supplement 1*. In the simulation, nuclear volume was assumed to be proportional to cell volume and

the initial nuclear volume was set as 2.9 fL (*Jorgensen et al., 2007*). Cln3 concentration equals to Cln3 protein number divided by nuclear volume. *Instantaneous_threshold* was set as 1.86 nM, and *Integration_threshold* was set as 10.6 nM*min. Other equations and parameters used in the simulation were kept the same as *Figure 2*.

## ODE model of the Start network

An explicit model of the whole Start network was constructed. Simulation results in, *Figure 3—figure supplement 1A–E* were produced by this model. Multiple phosphorylation of Whi5 was taken into account as well as the positive feedback loop. The equations and parameters are listed in *Supplementary file 1B*.

## Acknowledgements

We thank Y Tian for help with microfluidic device; the UCSF NIC Imaging Center for help with microscopy; R Dickson, M Funk, J Li, W Lim, A Miyawaki and J Weissman for plasmid; D Morgan for strain; Z Liu, S Su, Z Wang, and M Zhang for plasmid and strain construction; J Ferrell, H Lee, W Lim, Y Lu, M Kirschner, A Murray, S Oh, J Skotheim, and members of the laboratory for comments. The research was supported by NIH (R01 GM097115; P50 GM081879), NSF (DMR-0804183; CMMI-0941355), MOST (2009CB918500; 2012AA02A702; 2015CB910300), and National Natural Science Foundation of China (10721403; 11074009; 11721463; 11434001). XL and XY acknowledge the support from Lui Fellowship and Li Foundation. XL acknowledges the support from China Scholarship. SL acknowledges the support from National Natural Science Foundation of China (21103098).

## Additional information

### Funding

| Funder | Grant reference number | Author |
| --- | --- | --- |
| National Institutes of Health | R01 GM097115; P50 GM081879 | Chao Tang |
| National Science Foundation | DMR-0804183; CMMI-0941355 | Chao Tang |
| Ministry of Science and Technology of the People's Republic of China | 2009CB918500 | Chao Tang, Qi Ouyang |
| Ministry of Science and Technology of the People's Republic of China | 2012AA02A702 | Qi Ouyang |
| National Natural Science Foundation of China | 10721403 | Chao Tang |
| National Natural Science Foundation of China | 11074009; 11721463, 11434001 | Qi Ouyang |
| Lui Fellowship | Graduate Student Fellowship | Xili Liu, Xiaojing Yang |
| Li Foundation | Graduate Student Fellowship | Xili Liu, Xiaojing Yang |
| China Scholarship Council | China Scholarship | Xili Liu |
| Ministry of Science and Technology of the People's Republic of China | 2015CB910300 | Chao Tang |
| National Natural Science Foundation of China | 21103098 | Sen Liu |

The funders had no role in study design, data collection and interpretation, or the decision to submit the work for publication.

### Author contributions

XL, Conception and design, Acquisition of data, Analysis and interpretation of data, Drafting or revising the article; XW, Deducted the mathematical model, Performed the SSA and ODE simulations, Drafting or revising the article; XY, Helped with constructing the Cln3 plasmids, Conception and

design; SL, Modeled the 3D structure of Cln3-Cdk1 complex and performed the in silico alanine scanning; LJ, Performed the Whi5 dephosphorylation experiment; YQ, LH, Helped with constructing strains and plasmids; QO, Provided laboratory space, equipment and reagents, Supervised part of the work; CT, Conception and design, Drafting or revising the article

## Additional files

### Supplementary files

• Supplementary file 1. (**A**) Mathematic model of Whi5 kinetics. (**B**) ODE model of the Start network.

• Supplementary file 2. (**A**) Yeast Strain list. (**B**) Plasmid list.

### Major datasets

The following previously published datasets were used:

| Author(s) | Year | Dataset title | Dataset ID and/or URL | Database, license, and accessibility information |
|---|---|---|---|---|
| Bettayeb K, Oumata N, Echalier A, Ferandin Y, Endicott JA, Galons H, Meijer L | 2008 | Structure of phosphorylated Thr160 CDK2/cyclin A in complex with the inhibitor roscovitine | http://www.pdb.org/pdb/explore/explore.do?structureId=3DDQ | Publicly available at RCSB Protein Data Bank. |
| Takaki T, Echalier A, Brown NR, Hunt T, Endicott JA, Noble ME | 2009 | Crystal structure of CDK4/cyclin D3 | http://www.pdb.org/pdb/explore/explore.do?structureId=3G33 | Publicly available at RCSB Protein Data Bank. |
| Honda R, Lowe ED, Dubinina E, Skamnaki V, Cook A, Brown N, Johnson LN | 2005 | The structural basis of CDK2 activation by cyclin E | http://www.pdb.org/pdb/explore/explore.do?structureId=1W98 | Publicly available at RCSB Protein Data Bank. |
| Pevarello P, Brasca MG, Amici R, Orsini P, Traquandi G, Corti L, Piutti C, Sansonna P, Villa M, Pierce BS, Pulici M, Giordano P, Martina K, Fritzen EL, Nugent RA, Casale E, Cameron A, Ciomei M, Roletto F, Isacchi A, Fogliatto G, Pesenti E, Pastori W, Marsiglio A, Leach KL, Clare PM, Fiorentini F, Varasi M, Vulpetti A, Warpehoski MA | 2004 | Structure of CDK2/cyclin A with PNU-292137 | http://www.pdb.org/pdb/explore/explore.do?structureId=1VYW | Publicly available at RCSB Protein Data Bank. |
| Jeffrey PD, Russo AA, Polyak K, Gibbs E, Hurwitz J, Massague J, Pavletich NP | 1995 | Cyclin A-cyclin-dependent kinase 2 complex | http://www.pdb.org/pdb/explore/explore.do?structureId=1FIN | Publicly available at RCSB Protein Data Bank. |

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
