## [Decision Letter]

Thank you for sending your work entitled “Reliable cell cycle commitment in
budding yeast is ensured by signal integration” for consideration at
*eLife*. Your article has been favorably evaluated by Richard Losick
(Senior editor) and a member of the Board of Reviewing Editors (James Ferrell), and has
been reviewed in depth by three peer reviewers.

The Reviewing editor and the reviewers discussed their comments before we reached this
decision, and the Reviewing editor has assembled the following comments to help you
prepare a revised submission.

The manuscript “Reliable cell cycle commitment in budding yeast is ensured by
signal integration” by Liu et al. addresses the important problem of how yeast
cells make the decision to commit to a new round of cell division at the Start
transition. The authors reach the interesting conclusion that cells integrate Cln3
activity over time to make a reliable decision. This is in principle an important
finding and the integration model could be very general.

We have two main criticisms that need to be addressed prior to publication. There are a
number of minor criticisms as well, which would improve the rigor and understandability
of the paper.

Major concerns:

1) One major concern is that the integration model is not compared to alternative
models. I would suggest that the authors use statistical methods to prove that the
integration model provides a significantly better fit of the experimental data than
alternative models (for example instantaneous Cln3 concentration or instantaneous total
Cln3 in the whole cell). Notice that equation number 4 in the [Supplementary-material SD3-data] actually
suggests that the total amount of Cln3 in a cell should be a good estimator of when
cells divide. The authors need to provide stronger arguments to distinguish among
models. This should be readily doable, as they have already acquired the necessary
data.

2) An important assumption of the integrator model is that Whi5 has a
'memory', namely that its activity depends not only on the instantaneous Cln3
but also on previous events. This means that its dephosphorylation rate is slow. In the
supplementary materials, the authors cite published work from [8] that they say indicates that the half-time
for Cln3 dephosphorylation is longer than 30 min. However this is not exactly what
Charvin et al. said or what Charvin et al.'s data show.

What Charvin et al. showed (their Figure 4) is
that following a 15 min pulse of CLN2 expression in the presence of Clb-inhibitory
levels of Sic1-4A expression, Whi5 exits the nucleus and then begins to reenter it.
Re-entry begins very soon after the inducer of CLN2 expression is washed away and is
half-maximal by 35 min. Some of this time interval-Charvin et al. guess about 5-10
min-is probably due to the kinetics of CLN2 degradation. Presumably the remaining
∼25-30 min is partly attributable to the kinetics of Whi5 dephosphorylation, and
partly to the kinetics of Whi5 nuclear transport. So based on these experiments, one
might conclude that Whi5 dephosphorylation could have a half-time of up to 30 min, not
that it has a half-time of longer than 30 min.

There are two ways the authors could deal with this issue. They could directly measure
for themselves how fast Whi5 returns to the nucleus after inhibition of CDK1. One way
would be to use an analog-sensitive CDK1 mutant, which would obviate the need to guess
how long CLN2 remains above the level required for Whi5 phosphorylation after the
induction of CLN2 is terminated. While this might be the ideal solution, we appreciate
that there is already a lot of data in the paper, and we would be satisfied if the
authors would just accurately state Charvin et al.'s findings in the main text and
acknowledge that slow Whi5 dephosphorylation kinetics remains a key incompletely-tested
assumption of the authors' model.

---

## [Author Response]

*1) One major concern is that the integration model is not compared to
alternative models. I would suggest that the authors use statistical methods to prove
that the integration model provides a significantly better fit of the experimental
data than alternative models (for example instantaneous Cln3 concentration or
instantaneous total Cln3 in the whole cell). Notice that equation number 4 in
the*
[Supplementary-material SD3-data]
*actually suggests that the total amount of Cln3 in a cell should be a good
estimator of when cells divide. The authors need to provide stronger arguments to
distinguish among models. This should be readily doable, as they have already
acquired the necessary data*.

We acknowledge the referees’ suggestion.

When deducting equation number 4 in the [Supplementary-material SD3-data], we assumed Cln3 concentration is constant
during G1 phase. G1 length is inversely proportional to average Cln3 concentration,
which equals to instantaneous Cln3 concentration under this assumption. We clarified the
assumption in [Supplementary-material SD3-data].

Start is triggered when phosphorylated Whi5,
*Whi5*_*p*_, reaches the threshold
*Whi5*_*p*_*(G1)=Whi5*_*tot*_*-Whi5*_*c*_.
When considering the fluctuation of Cln3 concentration with time,Whi5p(t)=∫0tk⋅[Cln3(t')]exp(−p[phostot]⋅(t−t'))dt'.

G1 length cannot be estimated by the instantaneous Cln3 concentration at one time point
but by the Cln3 integration through the whole G1 phase. We added an additional section
in [Supplementary-material SD3-data]A
to discuss how G1 length is determined when relaxing the assumption of constant Cln3
concentration.

We tested the Instantaneous Model with the Cln3 profiles measured in experiment, which
are more meaningful than stochastic simulations. In our experimental setup, Cln3 level
during G1 in one cell fluctuates with about 20% CV (see Figure 5). If the Instantaneous Model were true, cells should
pass Start at or near the peak of Cln3 profile. However, we found that in near 80%
cells, the timing of Cln3 peak is different from the timing of Start (see Figure 5). Even when we relax the
timing requirement of the two events to be within 10 min, there are still over 50% cells
that failed the test. The result is even more significant when focusing on G1 lengths
longer than 30 min (see Figure 5). It is thus very unlikely that Start is triggered by the instantaneous Cln3
concentration. We added this result as Figure 2—figure supplement 2 and a corresponding explanation in the main text.
A major difference between the Instantaneous Model and the Integration Model is the
memory length. The two models are equivalent in the limit of zero memory. We estimated
the memory length to be over 10 min (Figure 2—figure supplement 3), comparable to the G1 length in daughter cells
(See the next response).Author response image 1.The Instantaneous Model fails with the Cln3 profiles measured in experiment.
(A) Schematic plot of the test. The test is considered a pass if the timing of
Start is near the timing of Cln3 peak by the specified tolerance value. The
Cln3 profiles are from the real data. Open circles denote the raw data; solid
lines are the smoothing splines. (B-C) Test failure percentage. Grey bars
indicate the failure percentage and red bars indicate the percentage of cells
whose Cln3 peak value is more than 20% larger than Cln3 at Start, for all cells
(B) and cells with G1 longer than 30 min (C).

*2) An important assumption of the integrator model is that Whi5 has a
'memory', namely that its activity depends not only on the instantaneous
Cln3 but also on previous events. This means that its dephosphorylation rate is slow.
In the supplementary materials, the authors cite published work from*
[8]
*that they say indicates that the half-time for Cln3 dephosphorylation is longer
than 30 min. However this is not exactly what Charvin et al. said or what Charvin et
al.'s data show*.

*What Charvin et al. showed (their*
Figure 4*) is that
following a 15 min pulse of CLN2 expression in the presence of Clb-inhibitory levels
of Sic1-4A expression, Whi5 exits the nucleus and then begins to reenter it. Re-entry
begins very soon after the inducer of CLN2 expression is washed away and is
half-maximal by 35 min. Some of this time interval-Charvin et al. guess about 5-10
min-is probably due to the kinetics of CLN2 degradation. Presumably the remaining
∼25-30 min is partly attributable to the kinetics of Whi5 dephosphorylation,
and partly to the kinetics of Whi5 nuclear transport. So based on these experiments,
one might conclude that Whi5 dephosphorylation could have a half-time of up to 30
min, not that it has a half-time of longer than 30 min*.

*There are two ways the authors could deal with this issue. They could directly
measure for themselves how fast Whi5 returns to the nucleus after inhibition of CDK1.
One way would be to use an analog-sensitive CDK1 mutant, which would obviate the need
to guess how long CLN2 remains above the level required for Whi5 phosphorylation
after the induction of CLN2 is terminated. While this might be the ideal solution, we
appreciate that there is already a lot of data in the paper, and we would be
satisfied if the authors would just accurately state Charvin et al.'s findings
in the main text and acknowledge that slow Whi5 dephosphorylation kinetics remains a
key incompletely-tested assumption of the authors' model*.

We acknowledge the referees’ correction for our interpretation of Charvin et
al.'s data. To better understand the integration mechanism, we measured Whi5
dephosphorylation kinetics as the referees suggested. As most Whi5 is dephosphorylated
and resides in the nucleus during G1 phase, we could not directly measure Whi5
dephosphorylation in G1. Thus we measured Whi5 nuclear entry right after G1 by
inhibiting Cdk1 activity with a strain bearing a *cdc28-as1* allele. The
average half time of Whi5 nuclear entry, which corresponds to the memory length in the
mathematical model, is 13.7 min in mother cells and 10.6 min in daughter cells,
respectively. Note that the average G1 length (from cytokinesis to Start) in daughter
cells is 13.6 min in SD medium, a memory length of about 10 min should be long enough
for the cell to take the advantage of integration.

We added this result as Figure 2—figure supplement 3, and the corresponding explanation in the main text.